# MDS-Net: Multi-Scale Depth Stratification 3D Object Detection from Monocular Images

**DOI:** 10.3390/s22166197

**Published:** 2022-08-18

**Authors:** Zhouzhen Xie, Yuying Song, Jingxuan Wu, Zecheng Li, Chunyi Song, Zhiwei Xu

**Affiliations:** 1Institute of Marine Electronic and Intelligent System, Ocean College, Zhejiang University, Zhoushan 316021, China; 2The Engineering Research Center of Oceanic Sensing Technology and Equipment, Ministry of Education, Zhoushan 316021, China; 3Donghai Lab, Zhoushan 316021, China

**Keywords:** autonomous driving, monocular image, 3D object detection, computer vision

## Abstract

Monocular 3D object detection is very challenging in autonomous driving due to the lack of depth information. This paper proposes a one-stage monocular 3D object detection network (MDS Net), which uses the anchor-free method to detect 3D objects in a per-pixel prediction. Firstly, a novel depth-based stratification structure is developed to improve the network’s ability of depth prediction, which exploits the mathematical relationship between the size and the depth in the image of an object based on the pinhole model. Secondly, a new angle loss function is developed to further improve both the accuracy of the angle prediction and the convergence speed of training. An optimized Soft-NMS is finally applied in the post-processing stage to adjust the confidence score of the candidate boxes. Experiment results on the KITTI benchmark demonstrate that the proposed MDS-Net outperforms the existing monocular 3D detection methods in both tasks of 3D detection and BEV detection while fulfilling real-time requirements.

## 1. Introduction

The 3D object detection is a fundamental function to enable complex and advanced autonomous driving tasks, such as object tracking and event detection. Nowadays, most 3D object detection algorithms use the LiDAR point cloud [1,2,3,4,5] to provide distance information. Recently, 3D detection based on point cloud [1,2,3,6,7,8] has developed rapidly. A common theme among SOTA point-based 3D detection methods is to project the point cloud into sets of 2D views. For example, AVOD [9] projects the 3D point cloud to the bird’s-eye view (BEV), and then fuses the features from BEV and image to predict 3D bounding boxes. Although the view-based method is time-saving, it destroys the information integrity of the point cloud. The voxel-based method [1] divides the point cloud into regular voxels, and then regresses the 3D bounding boxes by the 3D convolutional network. It preserves the shape information but usually suffers from high time complexity. The set-based method [2,3] represents the point cloud as a point set and uses a multi-layer perceptron (MLP) to learn features from the unordered set of points directly. Through doing this, it reduces the impact of the point cloud’s local irregularity. However, it still has the problem of mismatch between the regular grid and the point cloud structure. In summary, the laser is hard to apply in real-time for large amounts of data. Furthermore, the irregularity of the point cloud increases the difficulty of using convolutional operations to extract features. By contrast, feature extraction from the image can be realized using the convolutional neural network and the camera is cheaper. However, due to the lack of reliable depth information, 3D object detection based on monocular images is generally considered more challenging.

At present, with the popularity of convolutional neural networks, a multitude of 2D object detection networks have been raised. Generating proposals by the region proposal network (RPN) has been widely adopted by recent two-stage frameworks. The two-stage approach [10,11,12,13] first generates the region of interest (RoI) by the RPN and then regresses the object in RoI. The proposal region generation used in this kind of network leads to high time complexity. The one-stage approach [14,15,16] omits the proposed region generation and directly regresses the bounding box from the grids divided from the image. The one-stage detection is usually more efficient as compared to the two-stage one. However, it suffers a severe imbalance between positive and negative samples due to the lack of RPN, and could easily be hampered by a wrong-designed hyperparameter of the anchor. To solve this problem, FCOS [17] and FoveaBox [18] adopted the concept of anchor-free to solve 2D object detection in a per-pixel prediction. They avoid calculating the IoU between ground truth boxes and anchors, and through which improve the efficiency and the detection accuracy of the network.

Accurate prediction of the object’s depth and angle is a critical challenge in monocular 3D object detection. The existing monocular 3D object detection algorithms either generate pseudo point cloud [19,20,21] or directly process images to realize the detection of 3D objects [22,23,24,25,26]. The former uses a neural network to predict the depth map of the monocular image, and converts the depth map into pseudo point cloud by camera intrinsics, and then uses a point-cloud-based 3D detection network to regress the 3D boxes. For example, Xu [20] used a 2D image network to predict the RoI, then extracted image features and pseudo point cloud features in the RoI for 3D prediction. The algorithm does not sufficiently meet the real-time requirements owing to the time and space complexity generated by the pseudo-point cloud. Quite apart from the extra time complexity introduced by the depth map prediction, the accuracy of monocular depth estimation algorithms usually limits the prediction effects of such networks. The latter uses prior knowledge to establish the relationship between 2D and 3D objects, and then regresses the object’s 3D localization. Recently, deep neural networks directly processing the RGB image have demonstrated accurate results. Chen first proposed Mono3D [22] to predict the object based on the priori hypothesis of the ground plane. Chabot proposed Deep-MANTA [23], which uses the 3D CAD model to match the object’s key points to estimate 3D dimensions and orientation. M3D-RPN [25] uses prior statistics to initialize the 3D parameter and achieves 3D estimation by a monocular 3D region proposal network. MonoEF [27] predicts the camera extrinsic parameters by detecting vanishing point and horizon change, and then designs a converter to rectify perturbative features in the latent space. GUPNet [28] designs a GUP module to obtain the geometry-guided uncertainty of the inferred depth generally by three steps: it predicts the heat map and 2D box, then uses the ROI align to obtain the region of interest, and finally regresses the 3D box in the region of interest. MonoDTR [29] uses the transformer to regress objects. It designs the DFE module to learn depth-aware features with auxiliary supervision and uses the DTR module to integrate context and depth-aware features. We notice that the three networks all need to generate extra information and therefore suffer from extra processing time cost. The MonoDTR needs additional time to generate depth maps, MonoEF needs extra network for feature transformation, and GUPNet consumes time to generate regions of interest. UR3D [30] is the most similar work to ours, which predicts a coarse depth range and a depth offset for each FPN layer based on the observation of depth and scale’s statistical distribution, thus using a piece-wise linear curve to fit the nonlinear depth-scale curve. However, UR3D uses an additional depth map as the input and the network’s depth prediction is not based on a precise pinhole model. The algorithm does not effectively exploit the relationship between the object’s depth and image scale in detection and therefore obtains limited accuracy.

The existing algorithms suffer from the following limitations. First, the relationship between the depth of the object and image scale is not effectively exploited. Secondly, the small derivatives of angle loss make the network difficult to regress while focusing on the consistency of angle prediction and overlap. This is not considered in the angle prediction. Finally, the NMS algorithm causes a large number of reasonable candidate boxes to be dropped. Aiming at solving the above problems, this paper proposes an end-to-end one-stage 3D detection network based on a Multi-scale Depth-based Stratification structure, MDS-Net. The novelty of the MDS-Net primarily lies in the following three newly obtained features. Firstly, we propose a depth-based stratification structure derived from the conventional Feature Pyramid Network (FPN) [31]. By establishing mathematical models between the object’s depth and 2D scale, we assign each feature map of FPN to a different predictable depth range based on prior statistics and improve the depth perception of our network without adding additional time complexity. Secondly, we design a novel angle loss based on the consistency of IoU and angle to improve the prediction precision of boxes and the converge speed of training at the same time, which also solves the problem of the small derivatives of the loss making the network difficult to regress. Thirdly, we design a density-based Soft-NMS algorithm for post-processing. Since MDS-Net generates multiple boxes for the same object, we believe that objects are more likely at locations with a high density of predictions.

In summary, our contributions are as follows:We propose a one-stage monocular 3D object detection network, MDS-Net, based on a Multi-scale Depth-based Stratification structure, which can accurately predict the object’s localization in the 3D camera coordinate system from the monocular image in an end-to-end manner. The proposed MDS-Net achieves state-of-art performance on the KITTI benchmark for the monocular image of 3D object detection.We design a novel angle loss function to strengthen the network’s ability of angle prediction.We propose a density-based Soft-NMS method to improve the confidence of credible boxes.

## 2. The Proposed MDS-Net

This chapter comprises five sections: In Section 2.1, we introduce the overall architecture of our network. In Section 2.2, we introduce the depth-based stratification structure, which improves monocular depth estimation. In Section 2.3, the formulation to transform the network output to the 3D coordinate is presented. In Section 2.4, we expound on our loss function and propose a novel angle loss to improve angle prediction accuracy. Finally, in Section 2.5, we detail a density-based Soft-NMS algorithm that reasonably processes the prediction box and increases the recall.

### 2.1. Network Architecture

As shown in Figure 1, our object detection network consists of three parts: backbone, FPN [31] and detection head. Our network uses the Darknet53 [15] as the backbone to generate three feature maps, $D_{1},D_{2}$ and $D_{3}$, with downsampling ratios of 32, 16 and 8, respectively. We use the Shape-Aware Convolution (SAC) to enhance the network’s perception of objects with different aspect ratios between the backbone and FPN. We then use the FPN to fuse features from different layers and obtain the fused feature maps, $F_{1},F_{2}$ and $F_{3}$. Finally, each feature map $F_{i}(i=1,2$, 3) is connected to two detection heads, $H_{k}(k=1$, 2). The detection head is composed of two branches. One branch is responsible for predicting the 3D parameters of the bounding box, the other is responsible for predicting the confidence, intersection-over-union(IoU), and center-ness of the object. During the inference, the predicted-IoU and center-ness are multiplied with the corresponding confidence to calculate the final score.

### 2.2. Depth Stratification

As shown in Figure 2, we assume that the course angle is 0, then according to the pinhole model, the relationship between 3D depth $Z_{3d}$ and 2D size $\left(w_{2d},h_{2d}\right)$ of the object can be approximatively calculated as follows:(1)$$\left\{\begin{array}{c} \frac{Z_{3d}-\frac{L_{3d}}{2}}{f_{u}}=\frac{W_{3d}}{w_{2d}}\\ \frac{Z_{3d}-\frac{L_{3d}}{2}}{f_{v}}=\frac{H_{3d}}{h_{2d}}, \end{array}\right.$$
where $W_{3d},L_{3d}$ and $H_{3d}$ are the 3D sizes of the object, and $f_{u}$ and $f_{v}$ are the camera’s focal length. It can be inferred from Equation (Equation 1) that when focusing on objects of the same class that have similar 3D sizes, 2D sizes are mainly determined by depth. Based on this deduction and inspired by [31], which uses the Feature Pyramid Network (FPN) to detect objects of different scales, we apply the FPN structure to further improve the depth prediction by capturing objects’ size gap on the image, as shown in Figure 1. The feature maps’ output from the FPN structure are denoted as $F_{1},F_{2}$ and $F_{3}$, respectively, in ascending order of resolution, and each feature map is connected to two detection heads. The depth ranges predicted by these six detection heads are $\left[\varphi \cdot 2^{i+k-2},\varphi \cdot 2^{i+k-1}\right]$, respectively, where φ is a hyperparameter used to control the predictable ranges. During training, we assign the ground truth boxes to the reasonable detection heads based on the predefined predictable depth range, and each head is only responsible for predicting objects in the corresponding depth range.

Our proposed depth-based stratification structure assigns the long-distance objects to a high-resolution feature map following the pinhole model to obtain precise depth prediction. The depth-based stratification structure has the following two superiorities. Firstly, each feature map’s predictable depth range is increased twice as much as the previous layer. It is wise to keep the growth rate of the feature map’s predictable depth range consistent with the receptive fields for a better network depth prediction. Secondly, the three feature maps’ predictable depth ranges actually overlap each other for two reasons. On the one hand, from (Equation 1), we can deduce that the 2D size of the object is not strictly inversely proportional to the depth but is also affected by the object’s pose. We need to expand each feature map’s predictable depth range to strengthen the network’s prediction robustness. On the other hand, our overlapping stratification structure enables the network to predict objects in crucial areas multiple times, which in consequence eases the imbalance between positive and negative samples.

To further verify the proposed model Equation (Equation 1), we plot the correlation of 3D depth and 2D scale for all ground-truth boxes in the KITTI dataset, which are shown as the three colored scatters in Figure 3a,b, and produce the fitting curve of Equation (Equation 1) by replacing the $W_{3d},L_{3d},H_{3d}$ with the statistical average, which is shown as the red curves in Figure 3a,b. It should be noted that the practical correlation between $Z_{3d}$ and $h_{2d}$ (shown in Figure 3a) fits with the proposed model Equation (Equation 1) much better than the $w_{2d}$ (shown in Figure 3b). For the $w_{2d}$, the pose impacts the object’s 2D width on the image and the 3D width of each category varies widely. By contrast, the $h_{2d}$ is less affected by pose and has smaller inter-class variance and intra-class variance.

Based on this observation, we propose a more adaptive convolutional structure—Shape-Aware Conv (SAC), which is more robust to objects of various scales, as shown in Figure 1. The SAC performs both 1×3 and 3×1 convolutions on the input feature maps in parallel and then concatenates the two outputs with the input feature map followed by a 1×1 convolution to yield the final output feature map. As shown in Figure 4, the statistical distribution of the 2D aspect ratios in the KITTI dataset indicates that the aspect ratios of most objects are distributed between 1:3 and 3:1. Therefore, the network with SAC is more compatible with various aspect ratios of objects and thus improves the network’s robustness by solving the underfitting problem caused by the depth stratification structure.

### 2.3. Network Predictions

The outputs of our network consist of two parts. The first part is the predicted 3D parameters of an object, including its 3D center, 3D size, and angle. The second part is the predicted score of an object, including its confidence, predicted-IoU, and center-ness.

#### 2.3.1. 3D Prediction

According to the pinhole model of the camera, we project the object’s 3D center ${\left[X_{3d},Y_{3d},Z_{3d}\right]}^{T}$ in the camera coordinate system into the image.
(2)$$\left|\begin{array}{ccc} f_{u} & 0 & c_{u}\\ 0 & f_{v} & c_{v}\\ 0 & 0 & 1 \end{array}\right|\times \left|\begin{array}{c} X_{3d}\\ Y_{3d}\\ Z_{3d} \end{array}\right|=Z_{3d}\left|\begin{array}{c} u\\ v\\ 1 \end{array}\right|,$$
where *u* and *v* denote the coordinates of the projected center in the image. $f_{u}$ and $f_{v}$ are the focal length and $c_{u}$ and $c_{v}$ are the principal point offset of the camera.

According to the formula Equation (Equation 2), we can obtain the relationship between u,v and $X_{3d},Y_{3d},Z_{3d}$ as follows:(3)$$\left\{\begin{array}{c} X_{3d}=\frac{\left(u-c_{u}\right)Z_{3d}}{f_{u}}\\ Y_{3d}=\frac{\left(v-c_{v}\right)Z_{3d}}{f_{v}}, \end{array}\right.$$
where $c_{u},c_{v},f_{u},f_{v}$ are the parameters of the camera, and $u,v,Z_{3d}$ can be predicted by the network.

For the 3D size prediction, we calculate the average 3D size $\left(w_{0},l_{0},h_{0}\right)$ of all objects in the data set as the preset value and then predict the logarithm of the ratio between the ground truth and the preset value. We use the observation angle, which is intuitively more meaningful when processing image features, in the angle prediction. Following Kinematic 3D [32], we split the observation angle β into the object heading ${\;\theta }_{h}$, axis α, and angle offset θ in Figure 5.

Suppose the predicted values on the feature map $F_{i}$ are $\hat{u},\hat{v},\hat{Z},\hat{w},\hat{l},\hat{h},\hat{\alpha },\hat{\theta },{\hat{\theta }}_{h}$, then we use the following formula to calculate the 3D parameters $X_{3d},Y_{3d},Z_{3d},W_{3d},L_{3d},H_{3d},\beta$ of the object:(4)$$\begin{array}{cc} \; & \left\{\begin{array}{cc} Z_{3d} & =\varphi \cdot 2^{i+k+\hat{Z}-2}\\ X_{3d} & =\frac{\left[X_{P}+\hat{u}-c_{u}\right]\left(\varphi \cdot 2^{i+k+\hat{Z}-2}\right)}{f_{u}}\\ Y_{3d} & =\frac{\left[Y_{P}+\hat{v}-c_{v}\right]\left(\varphi \cdot 2^{i+k+\hat{Z}-2}\right)}{f_{v}}, \end{array}\right.\\ \; & \left\{\begin{array}{cc} W_{3d} & =e^{\hat{w}}\cdot w_{0}\\ L_{3d} & =e^{\hat{l}}\cdot l_{0}\\ H_{3d} & =e^{\hat{h}}\cdot h_{0}, \end{array}\right.\\ \; & \beta =\left\{\begin{array}{c} \hat{\theta }-\pi ,if\;\;\hat{\alpha }=1,{\hat{\theta }}_{h}=1\\ \hat{\theta },if\;\;\hat{\alpha }=0,{\hat{\theta }}_{h}=0\\ -\hat{\theta },if\;\;\hat{\alpha }=1,{\hat{\theta }}_{h}=0\\ -\hat{\theta }+\pi ,if\;\;\hat{\alpha }=0,{\hat{\theta }}_{h}=1, \end{array}\right. \end{array}$$
where $\left(X_{P},Y_{P}\right)$ represents the center of the grid.

#### 2.3.2. Score Prediction

For the confidence prediction, according to KITTI’s standard [33] for difficulty level, we piecewise predict the confidence of different difficulty levels and set the ground truth to 1.0,0.8,0.6, and 0.4, respectively. Following FCOS [17], the center-ness is defined as the normalized distance from the grid’s center to the object’s 2D center. Following IoUNET [34], the IoU is defined as the intersection over the union of the 3D prediction box and the 3D ground-truth box.

### 2.4. Loss

For the positive samples’ assignment strategy, M3D-RPN [25] sets the girds containing the object center as positive samples. FCOS3D [35] sets the grids near the object center as positive samples. We set all the grids inside the projected ground-truth box as positive samples and those outside as negative to ease the imbalance between positive and negative samples. Considering that our network uses the depth-based stratification structure, we set the confidence label of a grid as “ignore” if its center lies within an object beyond the predictable depth range of the relative detection head. As the solution to the label ambiguity that occurs when two or more boxes overlap on the image, we set the grid only responsible for the closest object due to the visibility from the camera’s perspective. As shown in Figure 6, compared with M3D-RPN and FCOS3D, our network has more positive samples to ease the imbalance between positive and negative samples.

The loss *L* in our network is composed of classification loss $L_{c}$ and 3D box $L_{3d}$:(5)$$L=L_{c}+L_{3d},$$

We use the sum of confidence loss, center-ness loss, and IoU loss as the classification loss, and ignore the contribution from all samples labeled as “ignore”. Inspired by FCOS [17], we add two branches parallel with the confidence branch to predict the center-ness and IoU and use the Quality Focal Loss (QFL) [36] to optimize the confidence, center-ness, and predicted-IoU.

We add up localization loss, size loss, and angle loss as our $L_{3D}$ loss. We use the L2 loss to optimize the localization and size predictions. VoxelNet [1] directly uses the offset of radians as the loss function. However, in the case shown in Figure 7a, the overlap of the two boxes is considerable while the network still generates a large angle loss. To solve this problem, SECOND [37] proposed a new angle loss as follows:(6)$$L_{rot}=SmoothL1\left[sin\left(\beta -\hat{\beta }\right)\right],$$
where β and $\hat{\beta }$, respectively, denote the ground truth observation angle and the predicted observation angle. This function naturally models IoU against the angle offset function. However, in the case shown in Figure 7b, the derivative of the loss is improperly small despite the loss reaching the maximum value, making it difficult for the network to regress. To overcome the above problems, we design a new angle loss:(7)$$L_{rot}={\left(\theta -\hat{\theta }\right)}^{2}+{\left(\alpha -\hat{\alpha }\right)}^{2}\cdot sin\left(2\cdot \hat{\theta }\right)+{\left(\theta_{h}-{\hat{\theta }}_{h}\right)}^{2},$$
where α and $\hat{\alpha }$, respectively, denote ground truth axis and the predicted axis, θ and $\hat{\theta }$, respectively, denote ground truth angle offset and the predicted angle offset, $\theta_{h}$ and $\hat{\theta_{h}}$, respectively, denote the ground truth heading and the predicted heading. Our angle loss not only establishes a consistent model between the IoU and the observation angle but also makes it easier to train the network when the angle loss rises to the maximum.

We assume that the network has predicted an accurate $\hat{\theta }$. As shown in Figure 8 and Figure 9, the inaccurately predicted axis $\hat{\alpha }$ has a minor effect on the IoU when the ground truth angle offset is close to either $0^{\circ }$ or $90^{\circ }$, while it has a significant impact on the IoU when the ground truth angle is close to $45^{\circ }$. We apply a weighted parameter $sin(2\cdot \hat{\theta })$ to the axis loss to increase the penalty for the inaccurate axis prediction when the ground truth angle is close to $45^{\circ }$.

### 2.5. Density-Based Soft-NMS

As shown in Algorithm 1, the traditional Soft-NMS algorithm [38] selects the detection box $b_{i}$ according to the predicted confidence score in descending order. It uses the IoU-based weighting function $f\left(IoU\left(M,b_{i}\right)\right)$ to decay the confidence score of the box $b_{i}$ that has a high IoU with the set of the predicted boxes *M*, as follows:(8)$$f\left(IoU\left(M,b_{i}\right)\right)=e^{-\frac{IoU{\left(M,b_{i}\right)}^{2}}{\sigma }},$$
where σ is a hyperparameter of the decay coefficient.

**Algorithm 1:** The pseudo-code of density-based Soft-NMS.  **Input:**    $B=\left\{b_{1},\ldots ,b_{n}\right\},S=\left\{s_{1},\ldots ,s_{n}\right\},N_{t}$,    *B* is the list of initial detection boxes,    *S* contains corresponding detection scores,    $N_{t}$ is the NMS threshold,   1: D←{},   2: $B_{0}\leftarrow B$,   3: **while**
B≠empty
**do**   4:  *m* ← *argmax*
*S*
   5:  *M* ← *b_m_*   6:  *D* ← *D* ∪ *M*, *B* ← *B* − *M*,   7:  
**for**
*b_i_*
*in*
*B*
**do**   8:   
**if**
*IoU* (*M*,*b_i_*) ≥ *N_t_*
**then**   9:    
*s_i_* ← *s_i_* · *f* (*IoU*(*M*, *b_i_*))   10:   **end if**   11:  **end for**   12:  *s_m_*←*s_m_* · *g* (*IoU*(*b_m_*, *B*_0_))   13: **end while**   14: **return**
*D*, *S*

Since our network predicts redundant boxes for one object in different depth stratifications, we develop a density-based Soft-NMS algorithm to filter repeated boxes. The key strategy of our NMS algorithm is that the denser the predicted box is, the more likely the object exists. We define the density of a candidate box as the sum of IoU between the candidate box and all surrounding boxes. Based on the results obtained by the 3D Soft-NMS algorithm, we use the density-based weighting function $g\left(IoU\left(b_{m},B_{0}\right)\right)$ to activate the candidate box, which has a high density:(9)$$g\left(IoU\left(b_{m},B_{0}\right)\right)=2-e^{-\frac{\sum_{b\in B_{0}}IoU{\left(b,b_{m}\right)}^{2}}{\gamma }},$$
where γ is another hyperparameter of the decay coefficient.

## 3. Experimental Results

### 3.1. Dataset

We evaluate our MDS-Net on the challenging KITTI data set [33]. The 3D detection of this data set contains two core tasks: bird’s eye view (BEV) object detection and 3D object detection. The dataset assigns its samples to three difficulty categories (easy, medium, and difficult) according to the object’s truncation, occlusion, and 2D bounding box height. We evaluate our algorithm on KITTI’s three classes in this work. Following [21], we apply the two split methods of the validation dataset [39] and official test dataset [33]. In each split method, the data from the same sequence will only appear in one split so that the interference of adjacent frames to the network model is eliminated.

Following [40], KITTI uses Average Precision on 40 recall positions $\left(AP_{40}\right)$ on the official test dataset to replace Average Precision on 11 recall positions $\left(AP_{11}\right)$. On the official test dataset and validation dataset, we show the $AP_{40}$ with a 0.7 IoU threshold, unless otherwise noted.

### 3.2. Implementation Details

Seeing that the center-ness and IoU prediction need a better box prediction result, we first train the location regression network except for the center-ness and IoU branches with a piecewise decayed learning rate. The initial learning rate is $10^{-4}$, and it will decay to $3\cdot 10^{-5}$ and $10^{-5}$ at the 30th epoch and 50th epoch, respectively. We then train the score branch for 10 epochs with a learning rate of $10^{-5}$. We use the pre-trained weights under the COCO dataset [41] to initialize our backbone. In our density-based Soft-NMS algorithm, σ is a hyperparameter of the weighting function $f\left(IoU\left(M,b_{i}\right)\right)$, $N_{t}$ is the threshold of IoU, and γ is a hyperparameter of the weighting function $g\left(IoU\left(b_{m},B_{0}\right)\right)$. A smaller σ means that the confidence of the redundant boxes is decayed more. A smaller setting of $N_{t}$ means that more boxes are considered as redundant boxes. A smaller setting of γ means that the confidence of the box that has a high density is activated more. We use Bayesian optimization to adjust the parameters, σ,γ and $N_{t}$, on the validation set with the $AP_{3D}$ of the medium level as the optimization goal for the three classes, and show the process of adjusting the parameters and the effect of the parameters on the $AP_{3D}$ score in Section 3.4.1. We then apply the optimized parameters to the test dataset. We obtain $\sigma =0.9,\gamma =25,N_{t}=0.7$ for car, $\sigma =1.0,\gamma =32,N_{t}=0.4$ for pedestrian, and $\sigma =1.2,\gamma =30,N_{t}=0.4$ for cyclist. For parameters of the Depth Stratification structure, as described in Section 2.2, The depth ranges predicted by the six detection heads are $\left[\varphi \cdot 2^{i+k-2},\varphi \cdot 2^{i+k-1}\right]$, with *i* and *k*, respectively, representing the index of feature maps and that of detection heads, and φ representing a hyperparameter that is used to control the predictable ranges of our framework. We obtain the parameter φ by counting the depths of the three classes in the KITTI training set, and produce the histograms of the depth of the three classes in the KITTI training set in Figure 10. From the results, we learn that the depth of cars mainly distributes between 5 m–70 m, while the depth of pedestrians and cyclists mainly distributes between 2.5 m–55 m. Therefore, the φ equal to 5 m, 2.5 m, and 2.5 m, respectively, for car, pedestrian, and cyclist. Due to the limitations of the network model, the maximum depth predicted by the network for the objects is 40 m when φ is set to 2.5. This will cause the network to give up predicting pedestrians and cyclists between 40 m and 55 m, which respectively occupy about 2% of pedestrians and 8% of cyclists in KITTI. This is worthwhile, considering the facts that close objects produce more dangers in autonomous driving tasks. Thus, we set the initial depth φ=5 m for car, φ=2.5 m for pedestrian, and cyclist in KITTI. We set (5 m, 80 m) as the predictable depth range for car and (2.5 m, 40 m) for pedestrian and cyclist, ignoring the objects outside the range.

### 3.3. Evaluation Results

We evaluate our network for the BEV object detection task and the 3D object detection task, each under both validation [39] and the official test dataset [33]. In Table 1, we compare the evaluation results of our network with the existing state-of-the-art monocular 3D detection algorithms. Our network obtains remarkable results in the car detection of all three difficulty levels, especially the easy level, for both tasks. For instance, under the test data split with IoU≥0.7, we surpass the previous state-of-the-art 3D object detection approach by +5.25 on easy, +2.70 on moderate, and +1.73 on hard for car. Moreover, in Table 2, for pedestrian and cyclist, we also surpass the previous state-of-the-art 3D object detection approach by +5.76/+2.56 on easy, +3.61/+1.01 on moderate, and +3.12/+0.86 on hard. In addition, we compare Table 3 the performance between our method and the three methods of MonoEF, GUPNet, and MonoDTR. The three methods obtain similar performance with ours while suffering from higher time complexity. In Table 3, in the easy level, our method surpass MonoDTR by +2.31 on $AP_{3D}$ and +4.22 on $AP_{BEV}$. This indicates that our method has better detection performance for easy objects. In addition, in other difficulty levels, the results of our method are also close to other methods. It is worth noting that our method does not need to generate extra information. Thus, our method has the shortest inference time among the above methods.

We evaluate our network on car in different depth ranges (5–20 m, 10–40 m, 20–80 m) with IoU criteria of 0.7 and 0.5 on the KITTI validation set to prove the validity of the depth stratification. The improvement is defined as the percentage gain of our algorithm over M3D-RPN. In Table 4, our method outperforms M3D-RPN [25] with margins on AP of +52.29%/+15.16% in 5–20 m, +72.87%/+29.21% in 10–40 m, and +169.81%/+53.55% in 20–80 m for criteria of IoU≥0.7 and IoU≥0.5. In order to evaluate our network for the depth prediction, we denote the average difference in the depth between every ground truth object and its best-predicted box as the average depth estimation error and calculate the error in intervals of 10 meters in Figure 11. As shown in the result, the depth estimation error increases as the distance grows. Moreover, we achieve a more accurate depth estimation improvement in 10–40 m, indicating that our network is able to obtain better predictions in critical areas.

It must be noted that by regressing the 3D bounding boxes on the image directly, our approach reaches the operating speed of 20 FPS and better meets the real-time requirements of autonomous driving, as compared to the existing networks based on the pseudo point cloud.

As mentioned in Section 2 of this paper, the depth stratification strategy enhances the model’s prediction for depth, the assignment strategy of the positive sample enhances the prediction for occluded objects, and the new angle loss function enhances the prediction for angles. To verify the validity of the components, we visualize some detection results in Figure 12. As shown in the results, our method obtains a much higher accuracy than M3D-RPN in the prediction of depth and angle, and works more effectively in the prediction of the occluded objects, as demonstrated in the left bottom plot of the figure. Therefore, our method obtains the improvement in $AP_{3D}$ and $AP_{BEV}$. This also proves the effectiveness of our proposed depth stratification strategy, the assignment strategy of the positive sample, and the new angle loss. Moreover, in Figure 13, we visualize more 3D detection results on the KITTI validation set [39].

### 3.4. Ablation Study

For the ablation study, following [39], we divide the training samples of the KITTI data set into 3712 training samples and 3769 verification samples and then verify the network’s accuracy according to the $AP_{40}$ standard.

#### 3.4.1. Depth Stratification

We compare the object detection $AP_{40}$ with and without depth stratification on the car. In the test without depth stratification, by referring to the strategy of the YOLO v3 network [15], we use IoU between the 2D box of the object and the preset anchor as the basis for stratification. As shown in the first and second rows of Table 5, the network based on Multi-scale Depth Stratification (MDS) (the second row) achieves a significant gain of 6.35 over the baseline implementation (the first row) on 3D detection on moderate, which verifies the superiority of the MDS structure.

In addition, we also compare the impact of different depth stratification strategies on the results. First, we set each FPN layer’s predictable depth range of the depth stratification structure to be an ’equal interval and no overlap’. We modify the network, so that each feature layer is connected to one detection head. The depth ranges of the predicted car of $F_{1},F_{2}$ and $F_{3}$ are [5, 30] m, [30, 55] m and [55, 80] m, respectively. For the second experiment, we add a feature layer $F_{0}$ with the downsampling ratio of 64 to the network, and each feature layer is connected to one detection head, and the depth ranges of the predicted car of $F_{0},F_{1},F_{2}$ and $F_{3}$ are [5, 10] m, [10, 20] m, [20, 40] m and [40, 80] m, respectively, which is denoted as ‘exponentially increasing interval and no overlap’. Finally, the depth stratification structure we described in Section 2.2 is denoted as an ‘exponentially increasing interval with overlap’. In Figure 14, we calculated the $AP_{40}$ in different depth ranges and we find that our depth stratification structure has better $AP_{40}$ for cars in different depth ranges, especially within 10–40 m. This verifies that the depth stratification of ‘exponentially increasing interval and no overlap’ allows the network to better learn the unified representation of objects in different depth ranges, and improves the robustness of the network.

#### 3.4.2. Density-Based Soft-NMS and Piecewise Score

As shown in Figure 15, we use Bayesian optimization to select parameters $\left(\sigma ,\gamma ,N_{t}\right)$ for density-based Soft-NMS and demonstrate the effect of these three parameters on the $AP_{40}$. Moreover we demonstrate the values of three parameters for the different classes when the $AP_{40}$ is highest in Section 3.2. To better understand the effect of the density-based Soft-NMS and piecewise score, we ablate them by using a standard NMS algorithm according to M3D-RPN and by setting all object scores to 1 as the baseline implementation (the second row in Table 5). We observe that both components achieve a considerable gain in both 3D and BEV perspectives. The combination of the two components surpasses the baseline by +9.17 on the 3D detection at the moderate level.

#### 3.4.3. Assignment Strategy of Positive Samples

We replace the positive samples assignment strategy with the strategies proposed by M3D-RPN [25] and FCOS3D [35]. We calculate the positive and negative samples ratios using different strategies and find that our proposed strategy has a more balanced positive and negative samples ratio. Moreover, we compare the 3D and BEV detection AP to verify our method’s effectiveness. As shown in Table 6, our positive samples’ assignment strategy improves performance by +6.95/+2.86 in ${AP}_{BEV}$ and +7.49/+1.84 in ${AP}_{3D}$ on moderate over the M3D-RPN and FCOS3D. The results demonstrate that our proposed strategy can effectively ease the imbalance between positive and negative samples to improve the robustness of the network.

#### 3.4.4. Angle Loss

We compare the 3D & BEV detection AP and Average Heading Similarity (AHS) of our proposed angle loss with the angle loss of AVOD [9] and that of SECOND [37]. It demonstrates that our angle loss focuses more on the consistency of angle prediction and overlap, rather than struggling with direction recognition in challenging cases. As shown in Table 7, our angle loss improves performance by +3.94/+3.38 in ${AP}_{BEV}$ and +2.74/+2.71 in ${AP}_{3D}$ over the SECOND and AVOD on moderate. Moreover our angle loss improves performance by 3.07/2.42 in AHS on BEV object detection task over the SECOND and AVOD on moderate. We also observe from the training process in practice that our angle loss improves the convergence speed of the angle regression, suggesting that we mitigate the problem of extrema convergence in the SECOND.

#### 3.4.5. Shape-Aware Convolution

As shown in Table 8, we compare the impact of different SAC structures. We use our networks without the SAC module as structure A. In Figure 16b, we add a branch of 3×1 convolution on A as structure B. The added branch is used to increase the horizontal receptive field of the network. Moreover, in Figure 16c, we add a branch of 1×3 convolution on A as structure C. The added branch is used to increase the longitudinal receptive field of the network. Finally, our proposed SAC structure is referred to as Structure D. The result demonstrates that our proposed SAC structure surpasses A by +1.94/+1.89 on easy, +1.15/+1.03 on moderate, and +0.99/+1.08 on hard for car and pedestrian class. This result shows that 3×1 and 1×3 convolution together improve car and pedestrian prediction. Shape-aware convolution did not improve the result for cyclist. According to the statistical results in Figure 4, we consider that this is because the cyclist in KITTI is relatively rare, which makes the network insensitive to the cyclist.

### 3.5. Limitations

As shown in Figure 17, the visualization of the dataset demonstrates that our model is insensitive to some close objects, causing inaccurate prediction or even missed detection of thses objects. The visualization illustrates that most of these objects are truncated by the images. Our proposed depth stratification strategy uses the mathematical relationship between size and depth in the image of an object to enhance the prediction of depth. However, when the object is truncated by the image, the mathematical relationship we established does not hold anymore, causing inaccurate prediction or even missed detection of these objects. Although severely occluded objects are not required to be detected in the KITTI dataset, in autonomous driving tasks, inaccurate prediction or missed detection of close objects is very dangerous. For the detection of truncated targets, a new branch is added after the output of the pyramid network, using a strategy for truncated objects, which is responsible for the detection of truncated objects. This is also part of our future work.

## 4. Conclusions

In this work, we propose a one-stage anchor-free network, which realizes direct detection of 3D objects from monocular images by applying the newly proposed depth-based stratification structure, IoU-aware angle loss, and density-based Soft-NMS algorithm. The experimental results based on the KITTI dataset [33] have demonstrated that our network can significantly improve the accuracy and recall rate in both bird’s-eye view object detection tasks and 3D object detection tasks. However, our network suffers decreased detection accuracy for the objects truncated by the image. We will investigate the strategies that can be applied to the proposed network to enhance the prediction of close objects.

## Figures and Tables

**Figure 1 sensors-22-06197-f001:**
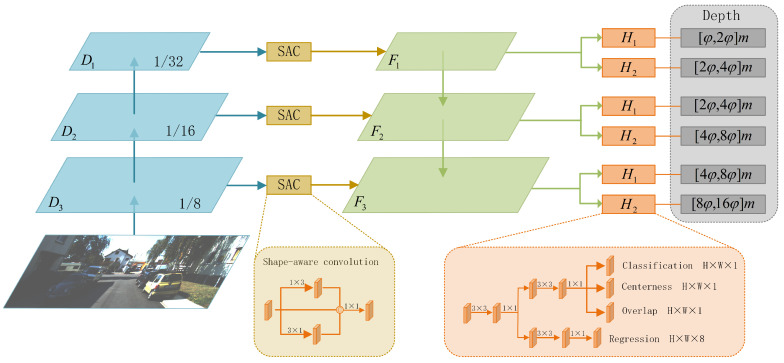
Overview of our proposed framework.

**Figure 2 sensors-22-06197-f002:**
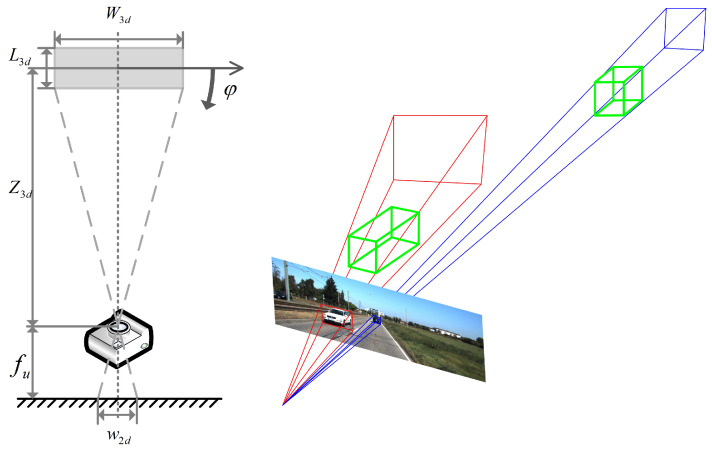
The relationship between the 3D depth and 2D scale of the object.

**Figure 3 sensors-22-06197-f003:**
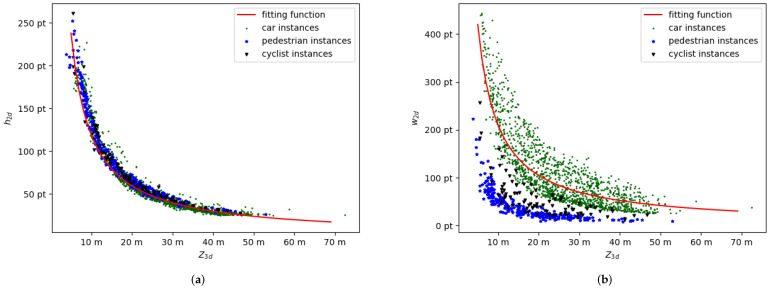
(**a**) The correlation of $Z_{3d}$ and $h_{2d}$ for part ground-truth boxes in the KITTI dataset and the fitting curve (red curve). (**b**) The correlation of $Z_{3d}$ and $w_{2d}$ for part ground-truth boxes in the KITTI dataset and the fitting curve (red curve). We use green points, blue stars, and black triangles to denote cars, pedestrians, and cyclists, respectively.

**Figure 4 sensors-22-06197-f004:**
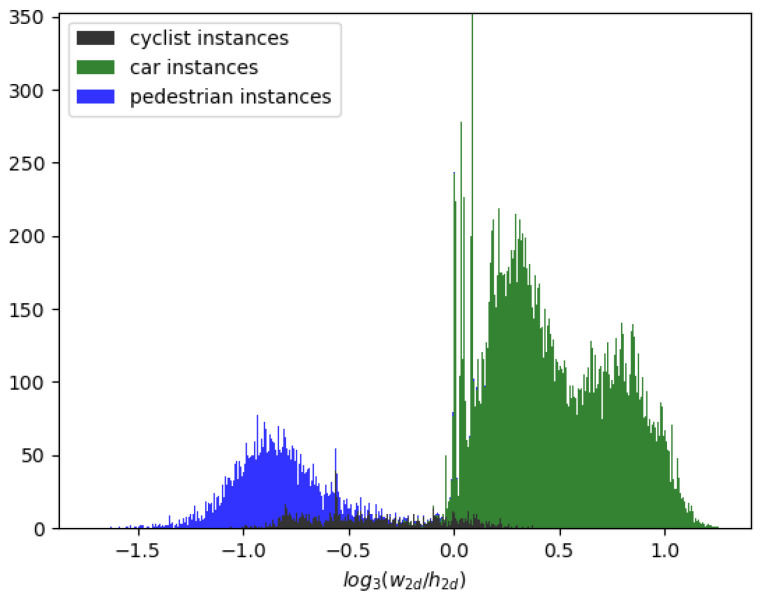
The statistical distribution of the 2D aspect ratios in the KITTI dataset.

**Figure 5 sensors-22-06197-f005:**
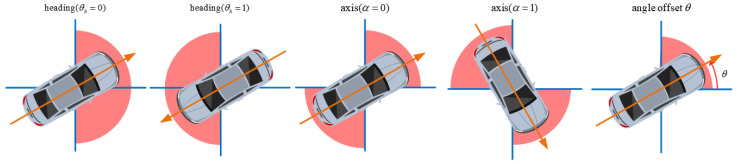
The decomposition of the observation angle β by the axis classification α, the heading classification $\theta_{h}$, and the angle offset θ.

**Figure 6 sensors-22-06197-f006:**
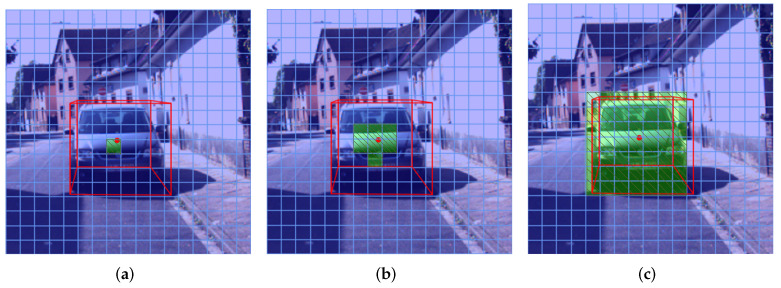
The comparison of positive samples’ assignment strategy among M3D-RPN (**a**), FCOS3D (**b**), and MDSNet (**c**). We use red box to denote the ground truth box, light blue grid to denote negative samples, and light green masked grid to denote positive samples.

**Figure 7 sensors-22-06197-f007:**
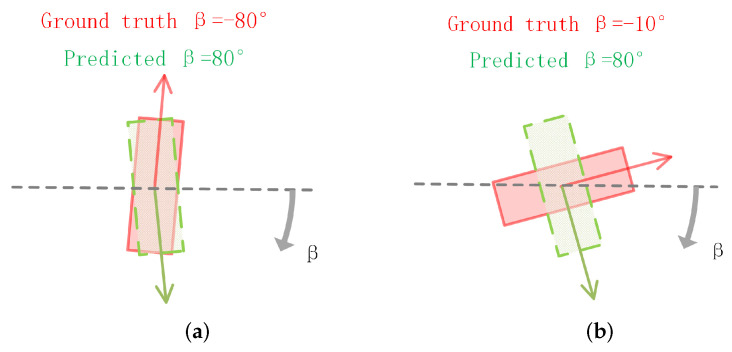
(**a**) The case where the angle loss function used in AVOD may encounter problems. (**b**) The case where the angle loss function used in SECOND may encounter problems. We use red solid box to denote the ground truth box and green-dotted box to denote the predicted box.

**Figure 8 sensors-22-06197-f008:**
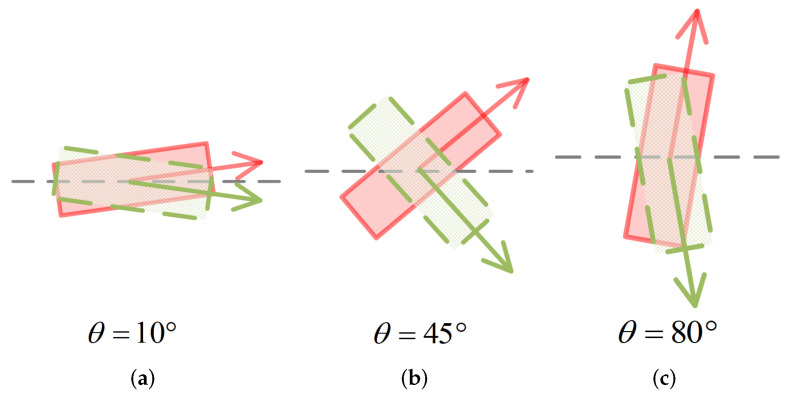
The influence of axis prediction error on IoU at different values of θ. We show the cases when θ = $10^{\circ }$ (**a**), θ = $45^{\circ }$ (**b**) and θ = $80^{\circ }$ (**c**). We use red solid box to denote the ground truth box and green-dotted box to denote the predicted box.

**Figure 9 sensors-22-06197-f009:**
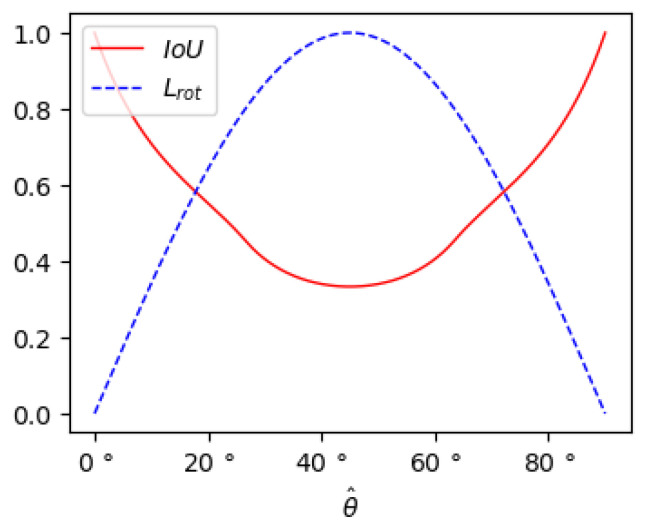
The curves of $L_{rot}$ and IoU with $\hat{\theta }$ when the network predicts an inaccurate axis $\hat{\alpha }$ and an accurate angle $\hat{\theta }$.

**Figure 10 sensors-22-06197-f010:**
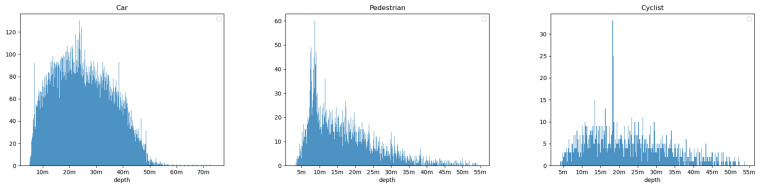
The histograms of the depth of the three classes in the KITTI training set.

**Figure 11 sensors-22-06197-f011:**
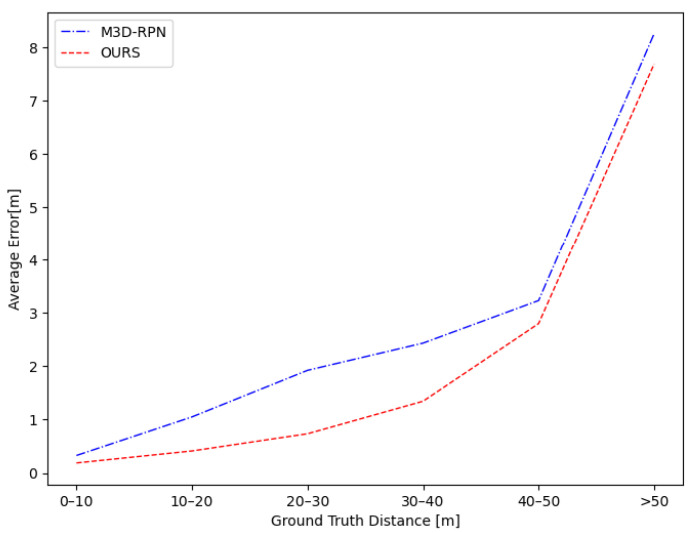
The average error of depth prediction visualized in different depth ranges.

**Figure 12 sensors-22-06197-f012:**
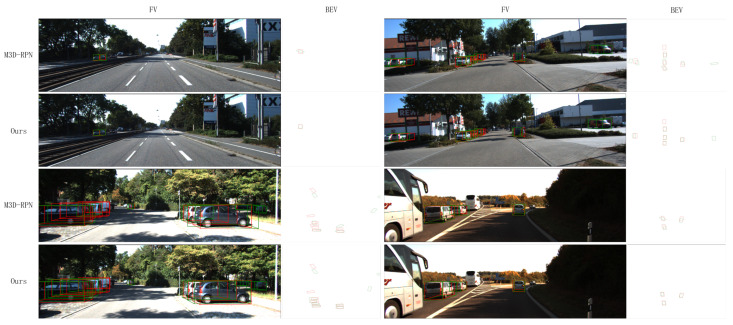
Qualitative results from our framework and MDSNet in the KITTI validation set. We use red box to denote the ground truth box and green box to denote the predicted box. It can be observed from the figure that our model has better accuracy for the prediction of depth and angle, and better performance for the detection of occluded objects.

**Figure 13 sensors-22-06197-f013:**
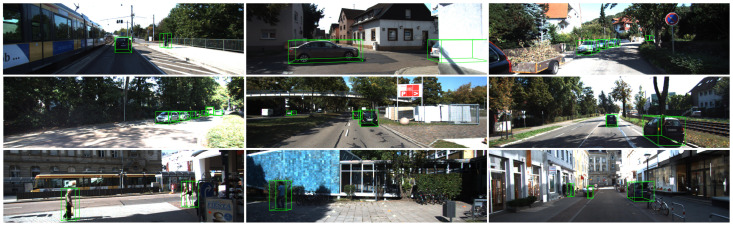
Qualitative results of our framework in the KITTI validation set.

**Figure 14 sensors-22-06197-f014:**
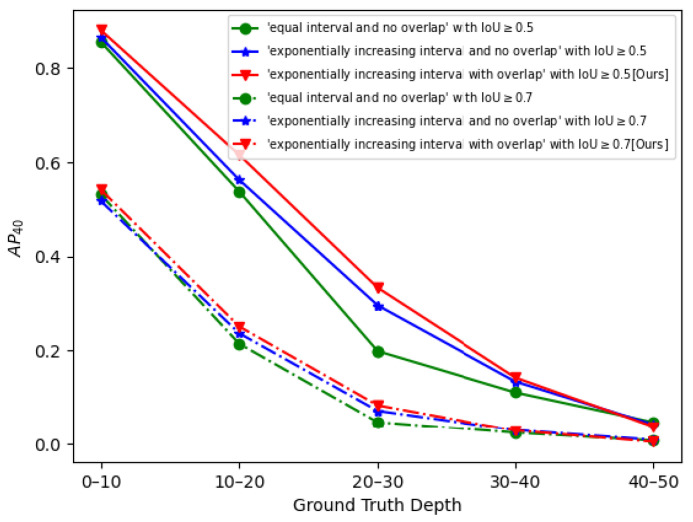
The ${AP}_{40}$ using different depth stratification models on the Bird’s Eye View object detection task in different depth ranges.

**Figure 15 sensors-22-06197-f015:**
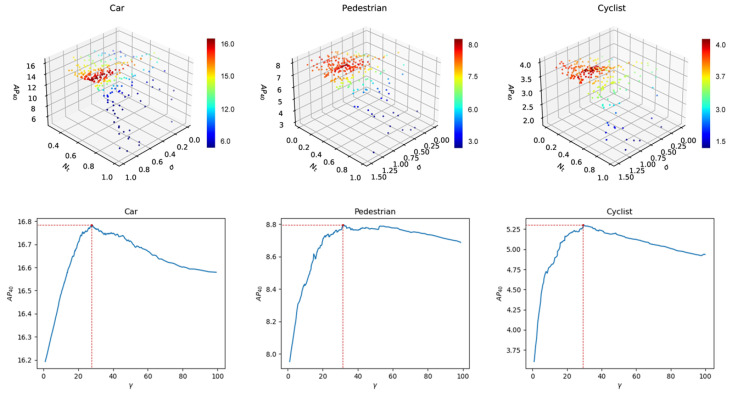
The effect of σ,γ and $N_{t}$ on the AP for the density-based soft NMS by applying Bayesian optimization. We first optimize σ and $N_{t}$ in the 3D soft NMS, and then optimize γ in the density-based Soft NMS.

**Figure 16 sensors-22-06197-f016:**
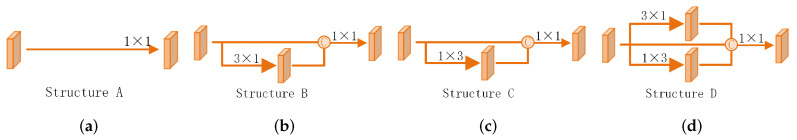
Different shape-aware convolution. We show the illustration of structure A (**a**), structure B (**b**), structure C (**c**), and structure D (**d**).

**Figure 17 sensors-22-06197-f017:**
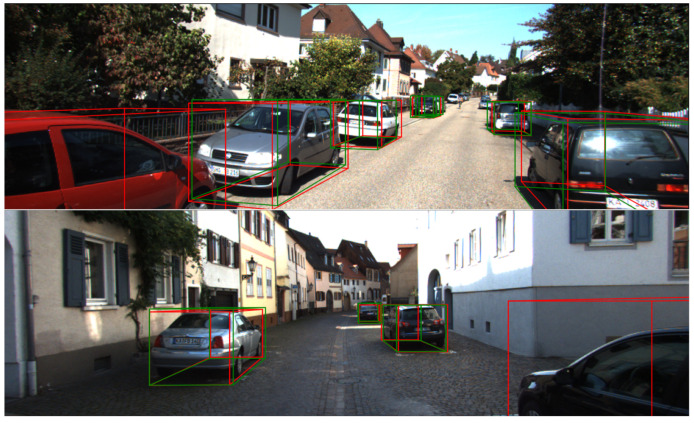
Some failure cases of our method. We use red box to denote the ground truth box, green box to denote the predicted box.

**Table 1 sensors-22-06197-t001:** The comparison of ${AP}_{40}$ between our framework and image-only 3D localization frameworks for car on the Bird’s Eye View object detection task and 3D object detection task. (The time is reported from the official leaderboard with slight variances in hardware). We use **bold type** to indicate the best result.

	Time	${AP}_{3D}$			${AP}_{BEV}$		
		Easy	Mod	Hard	Easy	Mod	Hard
M3D-RPN [25]	0.16 s	14.76	9.71	7.42	21.02	13.67	10.23
D4LCN [42]	0.2 s	16.65	11.72	9.51	22.51	16.02	12.55
UR3D [30]	0.12 s	15.58	8.61	6.00	21.85	12.51	9.20
PGD [43]	**0.03 s**	19.05	11.76	9.39	26.89	16.51	13.49
Ours	0.05 s	**24.30**	**14.46**	**11.12**	**32.81**	**20.14**	**15.77**

**Table 2 sensors-22-06197-t002:** The comparison of ${AP}_{40}$ between our framework and image-only 3D localization frameworks for pedestrian and cyclist on the 3D object detection task. We use **bold type** to indicate the best result.

	Time	Pedestrian			Cyclist		
		Easy	Mod	Hard	Easy	Mod	Hard
M3D-RPN [25]	0.16s	4.92	3.48	2.94	0.94	0.65	0.47
D4LCN [42]	0.2 s	4.55	3.42	2.83	2.45	1.67	1.36
PGD [43]	0.1 s	2.28	1.49	1.38	2.81	1.38	1.20
Ours	**0.05 s**	**10.68**	**7.09**	**6.06**	**5.37**	**2.68**	**2.22**

**Table 3 sensors-22-06197-t003:** The comparison of ${AP}_{40}$ between our framework and image-only 3D localization frameworks that require extra time complexity to generate additional information for car on the Bird’s Eye View object detection task and 3D object detection task.

	Time	${AP}_{3D}$			${AP}_{BEV}$		
		Easy	Mod	Hard	Easy	Mod	Hard
MonoEF [27]	-	21.29	13.87	11.71	29.03	19.7	17.26
GUPNet [28]	0.1 s	20.11	14.20	11.77	-	-	-
MonoDTR [29]	0.08 s	21.99	15.39	12.73	28.59	20.38	15.77
Ours	0.05 s	24.30	14.46	11.12	32.81	20.14	15.77

**Table 4 sensors-22-06197-t004:** The ${AP}_{40}$ for car in different depth ranges on the KITTI validation set. We use **bold type** to indicate the best result.

	IoU≥0.7			IoU≥0.5		
	5–20 m	10–40 m	20–80 m	5–20 m	10–40 m	20–80 m
M3D-RPN [25]	17.92	6.82	1.59	50.07	24.51	10.55
Ours	**27.29**	**11.79**	**4.30**	**57.67**	**31.66**	**16.20**
Improvement	+52.29%	+72.87%	+169.81%	+15.16%	+29.21%	+53.55%

**Table 5 sensors-22-06197-t005:** The ${AP}_{40}$ for ablation experiments on network components. We use **bold type** to indicate the best result.

Depth Stratification	Density-Based Soft-NMS	Piecewise Confidence	${AP}_{BEV}$			${AP}_{3D}$		
			Easy	Mod	Hard	Easy	Mod	Hard
			2.72	2.29	1.95	1.69	1.36	1.09
✓			14.99	12.32	10.86	9.83	7.71	6.84
✓	✓		20.69	17.09	15.16	14.67	12.03	10.43
✓		✓	30.99	20.09	16.42	21.35	13.6	10.78
✓	✓	✓	**34.56**	**22.86**	**18.56**	**25.30**	**16.88**	**13.53**

**Table 6 sensors-22-06197-t006:** The comparison of ${AP}_{40}$ among our positive samples’ assignment strategy, the strategy of M3D-RPN, the strategy of FCOS3D on the Bird’s Eye View object detection task, and 3D object detection task on the KITTI validation set. We use **bold type** to indicate the best result.

	${AP}_{BEV}$			${AP}_{3D}$			Positive:Negative (Samples)
	Easy	Mod	Hard	Easy	Mod	Hard	
M3D-RPN [25]	17.15	13.14	11.6	8.01	6.11	5.18	1:3716
FCOS3D [35]	24.34	17.23	14.84	17.62	11.76	9.93	1:429
Ours	**30.99**	**20.09**	**16.42**	**21.35**	**13.6**	**10.78**	**1:92**

**Table 7 sensors-22-06197-t007:** The comparison of the ${AP}_{40}$ and the AHS among our angle loss function, the angle loss function of M3D-RPN and the angle loss function of FCOS3D on the Bird’s Eye View object detection task and 3D object detection task on the KITTI validation set. We use **bold type** to indicate the best result.

	${AP}_{BEV}$			${AP}_{3D}$			AHS		
	Easy	Mod	Hard	Easy	Mod	Hard	Easy	Mod	Hard
AVOD [9]	24.78	16.71	13.71	16.87	10.89	8.77	16.67	11.36	9.44
SECOND [37]	25.74	16.15	12.91	17.83	10.86	8.46	16.97	10.71	8.62
Ours	**30.99**	**20.09**	**16.42**	**21.35**	**13.6**	**10.78**	**20.50**	**13.78**	**11.36**

**Table 8 sensors-22-06197-t008:** The comparison of the ${AP}_{40}$ among four shape-aware convolution structures in Figure 16 on the 3D object detection task on the KITTI validation set. We use **bold type** to indicate the best result.

	Car			Pedestrian			Cyclist		
	Easy	Mod	Hard	Easy	Mod	Hard	Easy	Mod	Hard
Structure A	19.41	12.45	9.79	8.42	6.52	5.18	**5.31**	**2.67**	**2.4**
Structure B	20.22	12.71	10.11	8.92	6.82	5.5	4.15	2.37	2.18
Structure C	20.16	12.81	10.17	9.03	6.9	5.67	4.55	2.06	1.87
Structure D	**21.35**	**13.6**	**10.78**	**10.31**	**7.55**	**6.26**	4.17	2.2	1.91

## Data Availability

The data sets presented in this study are available on http://www.cvlibs.net/datasets/kitti/, accessed on 29 June 2022. The codes presented in this study are available on request from the corresponding author. The code will be released in the future.

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
