# Peer review of "MDS-Net: Multi-Scale Depth Stratification 3D Object Detection from Monocular Images"

_sensors, 2022, doi:10.3390/s22166197_

Round 1

Reviewer 1 Report

The author propose a monocular 3D object detection method based on MDS-Net. The paper is generally intesting and useful. The following points can be improved:

1. The Introduction section must be improved. The state-of-the-art of the proposed method should presented detailedly. Other monocular 3D object detection methods and comparison with the proposed. What is advantage and disadvantage with other methods. And the necessity of the work should also be emphasized.

2. In section 4.2 (line 285-290), many parameters are set, the author should give some basis. How these parameters influence the final results. The author should also add some discussion and comparison.

3. The conclusion section must be enhanced. What is shortcome about this paper. And the author will do some future work should also be asdded.

Author Response

Thank you for your constructive comments on my manuscript entitled “MDS-Net: Multi-scale Depth Stratification 3D object detection from monocular images”(ID: sensors-1819486). Those comments are valuable and very helpful. We have discussed the comments carefully and based on which made corrections. The revisions are highlighted in the revised manuscript. Revision notes, point-to-point, are given in coverletter.

Reviewer 2 Report

This paper aims to improve the performance of monocular 3D object detection via depth-based stratification structure, IoU-aware angle loss, and density-based Soft-NMS algorithm. Overall,  the paper is well written and the proposed methods are easy to understand. However, the experiment part should be improved:

1. Some SOTA methods should be discussed and compared. For instance: 

[1] Kuan-Chih Huang, et al. MonoDTR: Monocular 3D Object Detection with Depth-Aware Transformer. 

[2] Yunsong Zhou, et al. Monocular 3d object detection: An extrinsic parameter free approach.

[3] Yan Lu, et al. Geometry uncertainty projection network for monocular 3d object detection.

2. Why the AP_3D and AP_BEV in Table 1 are quite improved, the authors should discuss the reasons in detail. For instance, visualizing some detection results from different methods is helpful. 

3. The limitation and failure cases of this method should be visualized and discussed. 

Author Response

(The authors gave the same response as above.)

Reviewer 3 Report

A model for multi-scale depth stratification 3D object detection from monocular images was proposed. The study content was relatively complete, different parts of the research and model were explained entirely, and I really liked this research. The manuscript is well prepared and appropriate for the journal.

Author Response

Thank you for your hard work. We are very grateful for your positive comments and encouragement to our work. We wish good health to you, your family, and your community. Thanks again for your hard work!

Round 2

Reviewer 1 Report

The author improved the paper in the revised manuscript. Accept!